# Synthetic hormone-responsive transcription factors can monitor and reprogram plant development

Arjun Khakhar[1†], Alexander R Leydon[2], Andrew C Lemmex[2], Eric Klavins[1]*, Jennifer L Nemhauser[2]*

[1]Department of Electrical Engineering, University of Washington, Seattle, United States; [2]Department of Biology, University of Washington, Seattle, United States

**Abstract** Developmental programs sculpt plant morphology to meet environmental challenges, and these same programs have been manipulated to increase agricultural productivity (Doebley et al., 1997; Khush, 2001). Hormones coordinate these programs, creating chemical circuitry (Vanstraelen and Benková, 2012) that has been represented in mathematical models (Refahi et al., 2016; Prusinkiewicz et al., 2009); however, model-guided engineering of plant morphology has been limited by a lack of tools (Parry et al., 2009; Voytas and Gao, 2014). Here, we introduce a novel set of synthetic and modular hormone activated Cas9-based repressors (HACRs) in *Arabidopsis thaliana* that respond to three hormones: auxin, gibberellins and jasmonates. We demonstrate that HACRs are sensitive to both exogenous hormone treatments and local differences in endogenous hormone levels associated with development. We further show that this capability can be leveraged to reprogram development in an agriculturally relevant manner by changing how the hormonal circuitry regulates target genes. By deploying a HACR to reparameterize the auxin-induced expression of the auxin transporter PIN-FORMED1 (PIN1), we decreased shoot branching and phyllotactic noise, as predicted by existing models (Refahi et al., 2016; Prusinkiewicz et al., 2009).

DOI: https://doi.org/10.7554/eLife.34702.001

*For correspondence:
klavins@uw.edu (EK);
jn7@uw.edu (JLN)

[†]These authors contributed equally to this work

**Competing interests:** The authors declare that no competing interests exist.

## Introduction

The body plans of plants are inherently plastic, making them amenable to optimization for a wide range of natural or artificial environments. Extrinsic and intrinsic cues are integrated by developmental programs to maximize the fitness of wild plants (*Vanstraelen and Benková, 2012*). Domestication of crops frequently relies on altering such programs to create more productive morphologies for agriculture, such as the dramatic reduction in bushiness of maize (*Doebley et al., 1997*) or the dwarfing of cereals that drove the green revolution (*Khush, 2001*).

Developmental programs are coordinated in large part by a set of hormones (*Vanstraelen and Benková, 2012*). Accumulation of a given hormone by de novo synthesis or transport influences the expression or activity of developmental master controller genes, analogous to wires in a circuit. Auxin, perhaps the best-studied hormone, controls many developmental programs that drive agriculturally relevant traits (*Weijers and Wagner, 20152016*). Many mathematical models connecting auxin signaling and transport at the molecular level to specific developmental phenotypes at the whole plant level have been developed (*Refahi et al., 2016*; *Prusinkiewicz et al., 2009*; *Smith et al., 2006*). These models highlight the importance of subtle parameters, like the strength of specific feedback loops in hormone signaling networks, in determining plant morphology.

While the ability of hormones to trigger and tune developmental programs makes altering hormonal signaling an attractive target for re-engineering the plant form, there are significant hurdles

**eLife digest** The genetic information of plants contains sets of instructions that shape a growing seedling. These 'developmental programs' are under the control of a range of hormones, such as auxin. Typically, the information from the hormones is relayed to the genetic material through proteins called transcription factors, which can act on DNA to turn specific genes on or off. Scientists have a good understanding of the roles of hormones, and they have created mathematical models that predict how changes in hormone levels affect the shape of a plant. However, it is still difficult to manipulate hormones inside a plant and test these models.

Here, Khakhar et al. created artificial transcription factors, referred to as HACRs, and put them into a plant called *Arabidopsis thaliana*. An HACR is made of different molecular modules stitched together. Each module has a precise role; for example, one turns off a specific gene, while another targets the HACR for destruction if a given hormone is present.

First, Khakhar et al. showed that HACRs could help track the levels of auxin in a developing plant. *Arabidopsis* plants were genetically engineered so that they would always produce a fluorescent protein. Then, an HACR was created that would switch off the gene for that fluorescent protein, so that no fluorescence would be present in the cell. If auxin was present, the HACR would get degraded, meaning fluorescence would appear. This helped to finely assess the amount of the hormone in various parts of the plant. By changing the modules in the HACRs, this approach could be applied to at least three other types of hormones.

Second, HACRs were used to reprogram *Arabidopsis* and change its appearance. For example, it is well known that auxin controls the number and location of branches on a plant. This complex process depends on how strongly auxin promotes the expression of a gene called *PIN1*. Khakhar et al. engineered an HACR that represses *PIN1*, and created a mathematical model that described the impact of this intervention. As predicted by the simulation, the HACR changed the strength of the relationship between *PIN1* and auxin, which resulted in plants with fewer branches – a trait that is of interest in farming.

HACRs are a new type of technology that is likely to work in a wide range of species. Ultimately, these artificial transcription factors could help to engineer plants that can face the disruptions brought by climate change, which would ensure better food security for people around the world.

DOI: https://doi.org/10.7554/eLife.34702.002

to overcome in such approaches. Native hormone signaling pathways are comprised of co-expressed and redundant components, embedded in highly reticulate cross-regulatory relationships with other signaling pathways, and have several layers of feedback (*Weijers and Wagner, 20152016*). For example, the auxin signaling pathway is comprised of three families of proteins, ARFs, AUX/IAAs, and TIR1/AFBs, all of which have multiple members with redundant regulatory roles and are cross regulated by a plethora of other signals (*Koltai, 2015*; *Naseem et al., 2015*).

Thus, there is a need for tools that can predictably alter how a specific hormone regulates a gene of interest to facilitate re-wiring plant development (*Brophy et al., 2017*). To date, such efforts have been largely limited to reducing or increasing expression of components of the native hormone signaling machinery (*Voytas and Gao, 2014*), an approach ill-suited for tuning the strength of connections within a network and easily confounded by redundancy and buffering within a network. In trying to circumvent redundancy, researchers are often forced to construct high order mutants of the multiple genes underlying the function of a single network hub. This approach reduces the precision of experimental or engineering interventions, as these genes are frequently only partially redundant with one another, and, thus this approach introduces more off-target effects. Chimeric promoters with altered hormonal regulation of a gene of interest have been used with some success (*Ulmasov, 1997*; *Rushton et al., 2002*). However, the paucity of detailed mechanistic maps connecting promoter architecture and chromatin state, and the high heterogeneity in these factors between genes, means that promoter design remains a bespoke approach with an associated high design and development cost for each network of interest. Additionally, these methods often require adding an extra copy of the gene of interest in a novel chromatin context, making it difficult to make definitive mechanistic conclusions. These challenges have made it difficult to study the significance

of hormone regulation on specific genes, particularly in regard to the impact of transcriptional feed-back loops on differentiation and morphogenesis. For all of these reasons, the potential predictive power of mathematical models has not been fully leveraged in the engineering of morphologies of agronomic interest. To facilitate more sophisticated interventions in plant developmental programs, we designed a set of synthetic and modular hormone-activated Cas9-based repressors (HACRs, pro-nounced 'hackers').

## Results and discussion

We previously validated the design of similar synthetic auxin-sensitive transcription factors in *Saccharomyces cerevisiae* (*Khakhar et al., 2016*). Guided by this work, we fused the deactivated Cas9 (dCas9) protein from *Streptococcus pyogenes* (*Gilbert et al., 2013*) to a highly sensitive auxin-induced degron (*Moss et al., 2015*) and the first 300 amino acids of the TOPLESS repressor (TPL) (*Pierre-Jerome et al., 2014*) (*Figure 1A*). The dCas9 associates with a guide RNA (gRNA) that targets the HACR to a promoter with sequence complementarity where it can repress transcription. Upon auxin accumulation, the degron sequence targets the HACR for ubiquitination and subsequent proteasomal degradation. Thus, in parallel to the natural auxin response, auxin triggers relief of repression on HACR target genes. Transgenic plants were generated with HACRs and a gRNA targeting a constitutively expressed Venus-Luciferase reporter, and, as expected, auxin treatment increased overall fluorescence (*Figure 1B,C*). A time-course using luciferase to quantify de-repression of the reporter supported these results with a significant spike in reporter signal (p<0.001, n = 10) peaking approximately 80 min post auxin exposure (*Figure 1D,E*). A HACR with a stabilized degron (*Moss et al., 2015*) showed significantly lower reporter signal upon auxin treatment (p=0.01, n = 10) (*Figure 1F*).

The modular nature of HACRs should allow substitution of the degron with any sequence that has a specific degradation cue. We tested this hypothesis by building HACR variants with degrons sensitive to two other plant hormones: jasmonates (JAs) (*Katsir et al., 2008*) and gibberellins (GAs) (*Murase et al., 2008*). Treatment of transgenic plants with exogenous hormones matched to the expressed variants significantly increased reporter signal as compared to control treatments (*Figure 1H,I,J*, *Figure 1—figure supplement 1*).

To rewire the connections between the hormone circuitry and developmental master controllers, HACRs must be able to respond to local differences in endogenous hormone levels. To visualize subtle differences in HACR sensitivity at the cellular level, we built a ratiometric auxin HACR by combining our previous design with a second reporter (tdTomato) driven by the same UBQ1 promoter driving the Venus reporter, with the only difference being that its gRNA target site was mutated (*Figure 2A*). An estimation of relative auxin levels was then calculated by normalizing the Venus reporter signal in each cell to that of the tdTomato signal in the same cell, minimizing any effect of differential expression of the UBQ1 promoter in different cell types. Using these lines, we visualized tissues at different developmental stages where auxin distributions had been previously described using auxin reporters like DII-VENUS or R2D2 (*Liao et al., 2015*). Auxin accumulation assayed by the HACR largely matched previous reports, such as the reverse fountain pattern of reporter signal in the root tip (*Band et al., 2014*) (*Figure 2B*) and higher signal in the vasculature as compared to the epidermis of the elongation zone (*Band et al., 2014*) (*Figure 2C*). We also observed high reporter signal in emerging lateral root primordia consistent with the auxin accumulation that triggers this developmental event (*Dubrovsky et al., 2008*) (*Figure 2D,E*).

To further explore the capacity of HACRs to respond to differences in endogenous hormone levels, we visualized the activity of auxin, GA and JA HACRs targeting a Venus reporter. Auxin accumulates in the apical domain of the early embryo and eventually resolves in later stages to the tips of the developing cotyledons, vasculature, and future root apical meristem (*Liao et al., 2015*)– the same patterns that were observed in plants expressing an auxin HACR (*Figure 2F–J*). In plants expressing a GA HACR, we observed a strong reporter signal in the early endosperm, consistent with the expression of GA biosynthesis enzymes (*Hu et al., 2008*) (*Figure 2K–M*, *Figure 2—figure supplement 1*). There are few reports of developmental regulation of JA distribution; however, we did detect accumulation of reporter signal in the developing ovule of plants expressing a JA HACR (*Figure 2—figure supplement 1*). Specifically, reporter signal appeared to be localized to the inner- and outermost layers of the integuments that surround the developing seed. We also observed that

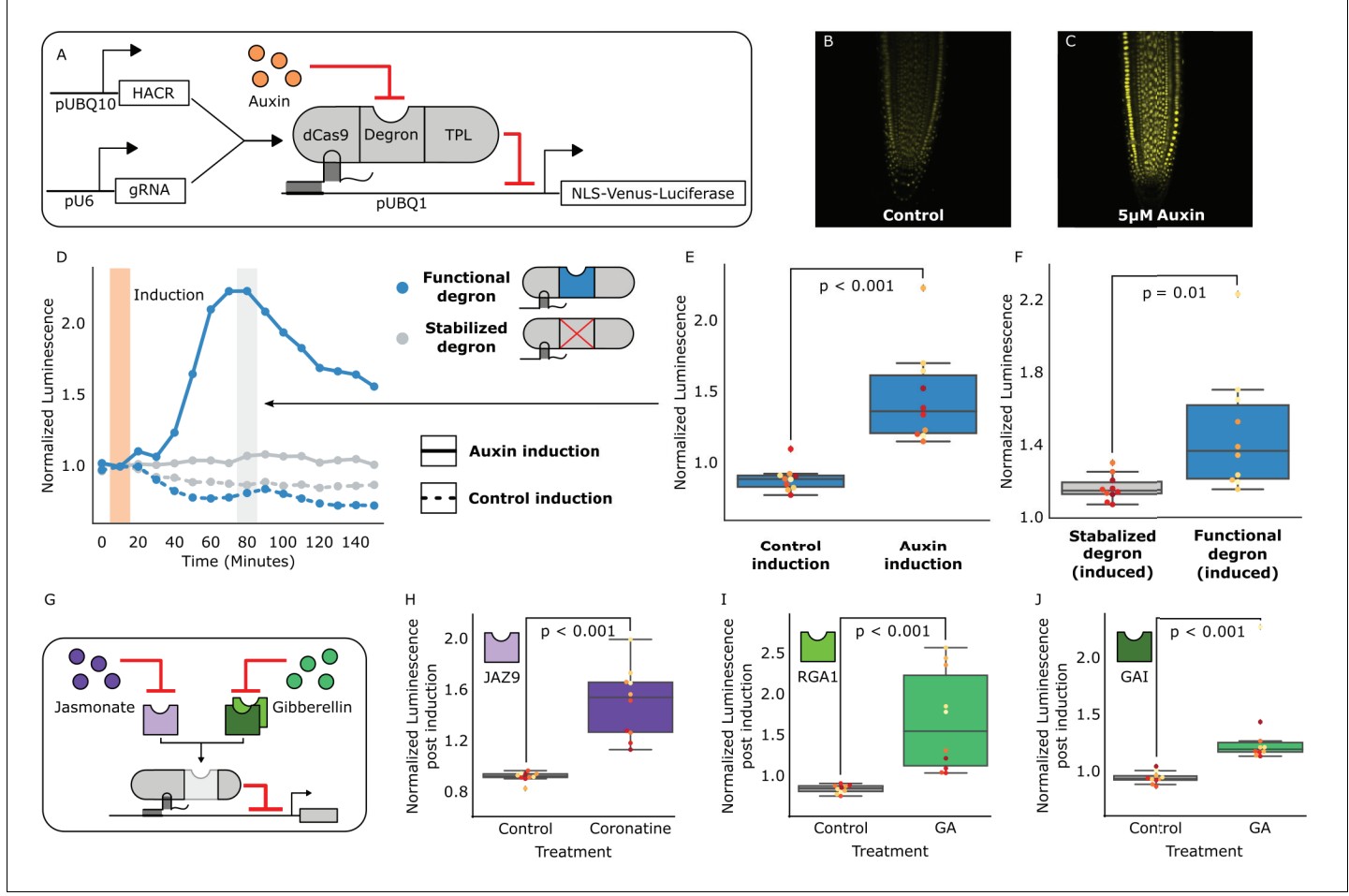

**Figure 1.** HACRs modulate gene expression upon exogenous hormone treatment. (**A**) A general schematic of the constructs transformed into *Arabidopsis thaliana* to test HACR hormone response. (**B,C**) Confocal microscopy images of root tips from plant lines with an auxin HACR regulating a Venus reporter 24 hr after treatment with (**B**) control or (**C**) 5 µM auxin. (**D**) An example of a luciferase based time course assay testing whole seedlings of an auxin HACR line treated with auxin (solid blue line) and a control (dashed blue line). The timepoint of auxin induction is highlighted with an orange bar. The time point of maximum auxin response is highlighted by the grey bar. (**E**) The difference between auxin and control induction at the time of maximum auxin response for the tested seedlings (n = 10) is summarized in the box plot. Every seedling is represented as a different colored dot. (**F**) A HACR variant line with a stabilized auxin degron was also assayed (D, solid and dashed grey lines) and the response to auxin of these seedlings compared to seedlings of the line with a functional auxin degron at the time of maximum auxin response are summarized in box plot in F. (**G**) A schematic of how the hormone specificity of HACRs were altered by swapping the hormone degron. (**H,I,J**) These box plots summarize the response of transgenic seedlings carrying these constructs (n = 10) to treatment with either control or the appropriate hormone. The degron used in the HACR is specified in the top left corner of the plot. Every seedling is represented as a different colored dot. All p-values reported were calculated using a one-way ANOVA.

DOI: https://doi.org/10.7554/eLife.34702.003

The following source data and figure supplement are available for figure 1:

**Source data 1.** Data for auxin HACR box plots in *Figure 1*.
DOI: https://doi.org/10.7554/eLife.34702.005
**Source data 2.** Data for auxin HACR time courses in *Figure 1*.
DOI: https://doi.org/10.7554/eLife.34702.006
**Source data 3.** GA HACR (PHD3) data for *Figure 1* and *Figure 1—figure supplement 1*.
DOI: https://doi.org/10.7554/eLife.34702.007
**Source data 4.** GA HACR (PHD6) data for *Figure 1* and *Figure 1—figure supplement 1*.
DOI: https://doi.org/10.7554/eLife.34702.008
**Source data 5.** JA HACR data for *Figure 1* and *Figure 1—figure supplement 1*.
DOI: https://doi.org/10.7554/eLife.34702.009

**Figure supplement 1.** The hormone specificity of HACR response can be predictably altered by including different hormone responsive sequences.

*Figure 1 continued*

DOI: https://doi.org/10.7554/eLife.34702.004

the JA HACR reporter was strongly induced in leaves subjected to mechanical damage (*Figure 2N–Q*), a condition known to induce high levels of JA (*Katsir et al., 2008*).

Beyond their application as sensors of endogenous hormone distributions, HACRs should also be capable of reprogramming how such signals are translated into plant morphology. To test this, we turned to shoot architecture, an agronomically important trait with a well-established connection to auxin. Fewer side-branches allow for higher density planting (*Khush, 2001*) and more regular arrangement of lateral organs (phyllotaxy) facilitates efficient mechanized harvest (*Burks et al., 2005*). The molecular mechanisms that control branching and phyllotaxy are well studied and have been mathematically modeled (*Refahi et al., 2016*; *Prusinkiewicz et al., 2009*). These models predict that a key parameter controlling both these processes is the strength with which auxin promotes its own polar transport (*Bennett et al., 2014*), which we will refer to as feedback strength. One molecular mechanism that contributes to this feedback is the auxin-induced increase in expression of the auxin transporter PIN-FORMED1 (PIN1) (*Vieten et al., 2005*). Thus far, it has been impossible to tune the strength of auxin-mediated transcriptional feedback on PIN1, and thus impossible to fully test its role in regulating shoot architecture or its potential for engineering this trait.

To test whether we could rationally alter shoot architecture by changing feedback strength, we generated transgenic plants with a HACR targeting PIN1 (*Figure 3A*), as well as a model that produced a qualitative hypothesis of the impact of this intervention (Supplementary note 1). Our model predicts that this perturbation will decrease the activation of expression of PIN1 by auxin and dampen the dose response relationship between auxin and PIN1 expression (*Figure 3—figure supplement 1B,C*). Quantitative PCR results on transgenic plants support these predictions, as the modest but significant reduction in PIN1 expression observed in plants expressing a PIN1 gRNA can be erased with exogenous auxin treatment (*Figure 3—figure supplement 1D*). Our model and these results highlight the substantial difference between regulation by a hormone-responsive transcription factor and a static repressor. Static repressors would consistently suppress target gene expression at all hormone levels. In contrast, HACRs dampen both the dynamic and steady state dose response relationship between hormone concentration and gene expression akin to modulating the gain in a circuit (*Figure 3—figure supplement 1B,C*).

In relation to shoot architecture models, the effect of an auxin-regulated HACR targeting PIN1 should be a reduction in feedback strength. In Prusinkiewicz et al. (*Prusinkiewicz et al., 2009*), auxin-regulated feedback is modeled as a post-translational mechanism dependent on the flux of auxin through the cell membrane. The magnitude of this flux is proportional to the recruitment of PIN1 to the membrane. According to their simulations, feedback strength is directly proportional to the number of branches the plant will develop. This effect is hypothesized to result from the reduced ability of lateral buds to establish auxin efflux into the main stem, an essential step in bud outgrowth (*Figure 3D*). While the transcriptional mode of feedback we are altering with our HACR is not directly encoded in the Prusinkiewicz et al. model, we hypothesized that decreasing transcriptional feedback strength would have qualitatively similar results to decreasing post-translational feedback strength. Thus, we expected a decrease in the number of branches in lines where auxin HACRs were targeted to PIN1. This is exactly what we observed (*Figure 3—figure supplements 2* and *5*). In lines with the strongest phenotypes, we observed roughly half the total number of branches per plant (*Figure 3E*). No difference in the number of branches was observed for lines that had a HACR with a stabilized auxin degron regulating PIN1 expression, suggesting this phenotype was not simply due to repression of PIN1 (*Figure 3—figure supplement 3*).

Feedback strength is also an important control parameter for the process of phyllotactic patterning. In the inhibition zone model, each primordium (*Figure 3F*, green circles) creates an inhibition zone around itself by depleting auxin (*Figure 3F*, shown in orange) from its surroundings, thereby preventing enough auxin to accumulate to form a new primordium. This zone is created by a feedback driven flow of auxin towards the primordium. The cells that are capable of forming new primordia are present in a region called the central zone periphery (*Figure 3F*, black ring) surrounding the shoot apical meristem (*Figure 3F*, green circle in the back ring). The overlapping inhibition zones

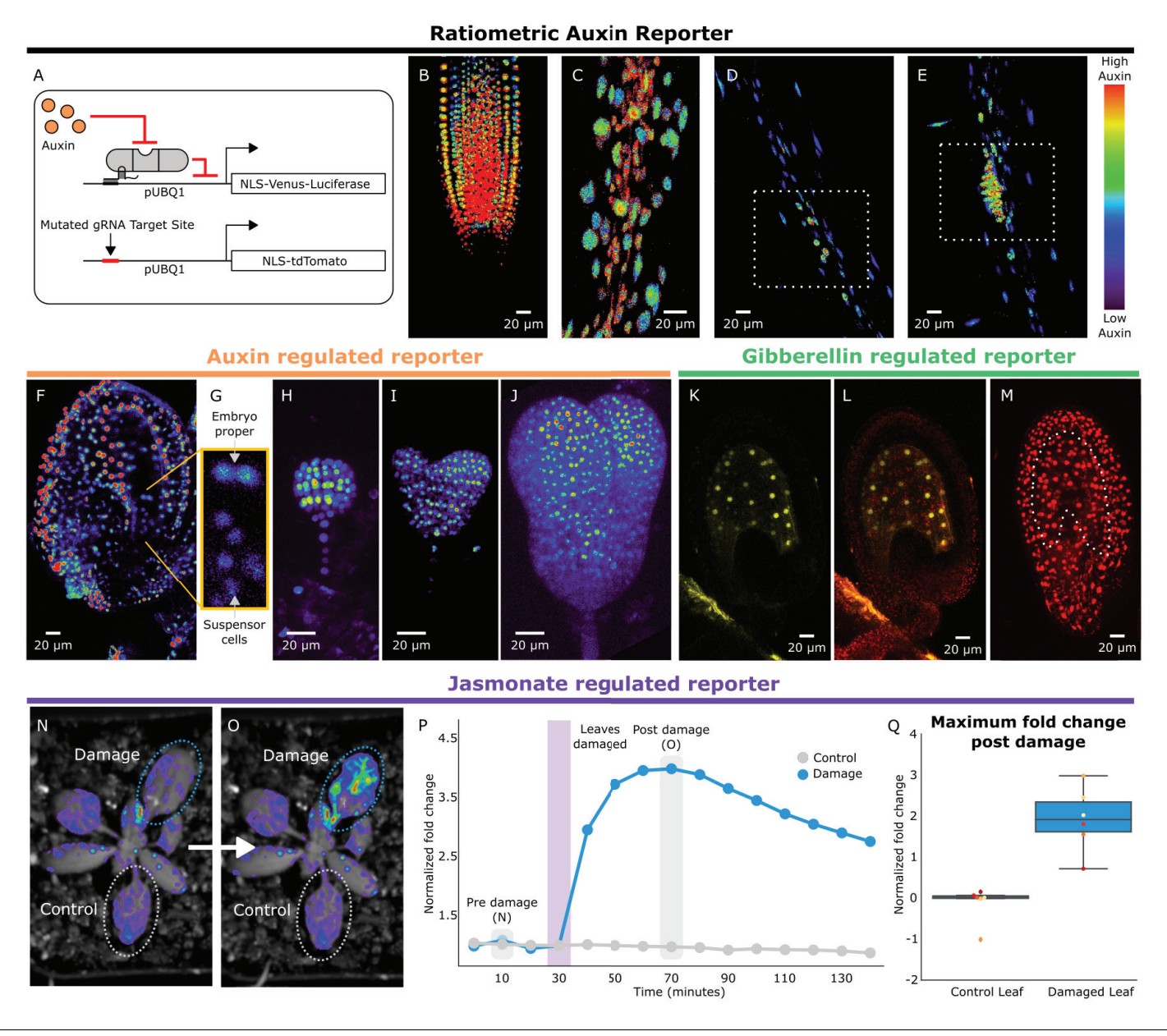

**Figure 2.** HACRs respond to endogenous hormone signals and can be used to study development. (**A**) Schematic of the genetic circuit used to build ratiometric lines of auxin responsive HACRs. In addition to an auxin HACR regulating a nuclear localized Venus-luciferase reporter the lines also have a nuclear localized tdTomato reporter being driven by a version of the UBQ1 promoter with the gRNA target site mutated. (**B–E**) Confocal microscopy images of roots of seedlings from lines described in A. Reporter signal in images is the background subtracted Venus signal normalized by the background subtracted tdTomato signal. Warmer colors correspond to higher normalized reporter signal. (**B**) The stereotypical reverse fountain pattern of auxin distribution is observed in the root tip. (**C**) Higher reporter signal is observed in the vasculature compared to the epidermis of the elongation zone of the root, consistent with auxin being trafficked along the vasculature. The dashed white boxes highlight high reporter signal in (**D**) the founder cells of lateral roots and in (**E**) a developing lateral root primordium. (**F–J**) Confocal microscopy images visualizing reporter signal of a non-ratiometric auxin HACR regulated reporter (**F**) in the ovule 48 hr post pollination, (**G**) in the two-cells embryo, (**H**) in the globular embryo, (**I**) in the heart stage embryo and (**J**) in the early torpedo stage embryo. Warmer colors correspond to higher reporter signal. (**K–M**) Confocal microscopy images visualizing reporter signal of a GA HACR regulated reporter (**K**) in the ovule 48 hr post pollination, (**L**) reporter signal merged with red auto-fluorescence to highlight the endosperm region and (**M**) an unregulated tdTomato reporter, with the endosperm highlighted with a dashed white line, for comparison. (**N–Q**) Visualization of JA HACR regulated reporter expression in leaves in response to mechanical damage using a luciferase-based assay. Images of leaves overlaid with the luciferase signal before (**N**) and after damage (**O**) are shown to the left of a representative plot of the normalized reporter signal over time (**P**). (**Q**) Box plot summarizing the maximum fold change at 70 min for control and damaged leaves. Points of the same color represent leaves from the same plant.

*Figure 2 continued on next page*

*Figure 2 continued*

DOI: https://doi.org/10.7554/eLife.34702.010

The following source data and figure supplement are available for figure 2:

**Source data 1.** Time course Damage assay data *Figure 2*.
DOI: https://doi.org/10.7554/eLife.34702.012
**Source data 2.** Boxplot Damage assay data *Figure 2*.
DOI: https://doi.org/10.7554/eLife.34702.013
**Figure supplement 1.** Distributions of auxin, gibberellin and jasmonate during early embryo development can be mapped out using HACRs.
DOI: https://doi.org/10.7554/eLife.34702.011

from all the existing nearby primordia leave only certain regions of the central zone periphery capable of forming new primordia (*Figure 3F*, dashed green circles on yellow arcs). A mathematical model by Refahi et al (*Refahi et al., 2016*). divides the central zone periphery into discrete units or cells and calculates a probability for each cell to form a new primordium at every timepoint. This probability is used to simulate the growth of the plant and estimate the expected frequency of phyllotactic patterning errors, such as co-initiation of primordia (*Figure 3F*, as shown in the grey meristem). This occurs when there is more than one region on the central zone periphery that is competent to form a primordia, leading to two primordia being initiated at the same time. According to the model, the radius of the inhibition zones is inversely proportional to the number of co-initiatiating primordia. In auxin HACR plants with a PIN1 gRNA, we hypothesized that lower feedback strength would lead to a less sharp auxin gradient around each primordium and thus a larger inhibition zone (*Bennett et al., 2014*) (*Figure 3F*, as shown in the blue meristem). Consistent with this prediction, plants with a HACR targeting *PIN1* showed a significant reduction in co-initiations (*Figure 3G*, *Figure 3—figure supplement 4*).

By making it possible to alter transcriptional feedback strength rather than simply gene expression, the HACR platform enabled exploration of previously inaccessible parameter regimes. This proof-of-concept establishes a new method for modifying a large number of desired traits. Additionally, the modular nature of HACRs allows for independent tuning of hormone sensitivity and repression strength (*Khakhar et al., 2016*), as well as allowing for tissue-specific modulation of target genes. These modifications could substantially extend the range of possible phenotypes and mitigate trade-offs, for example having few branches to fit more plants on a field versus the total number of fruits per plant. The use of HACRs here is among the first examples of utilizing synthetic signaling systems to re-engineer the morphology of a multicellular organism in a model-driven manner, a long standing goal across the fields of pattern formation and tissue engineering, and this strategy should be extensible to a wide variety of organisms, particularly given the success of implementing the auxin-induced degradation module (AID) in diverse eukaryotes (*Nishimura et al., 2009*). In agricultural settings, farmers already manipulate development or defense pathways by applying hormones or their synthetic mimics. HACRs could be used to connect these treatments with the expression of genes, such as those involved in defense, to create inducible traits. Additionally, HACRs could be extended to any other hormone that utilizes degradation-based signaling, such as salicyclic acid, strigalactones and karrikins. The wide range of degradation cues, the ease of targeting any gene, and the likely conserved function across angiosperms should mean that HACRs have the capacity to reprogram a plethora of developmental traits in a broad range of crop species.

## Materials and methods

### Construction of plasmids

Expression cassettes for the gRNAs, HACRs and the reporters were built using Gibson assembly (*Gibson et al., 2009*). These were then linearized by restriction enzyme digestion and assembled into a yeast artificial chromosome based plant transformation vector with kanamycin resistance using homologous recombination based assembly in yeast (*Shih et al., 2016*). The PIN1 gRNA expression vector and the additional tdTomato expression vector for the ratiometric lines were built using Golden-Gate assembly (*Engler et al., 2008*) into the pGRN backbone (*Hellens et al., 2000*) with hygromycin resistance.

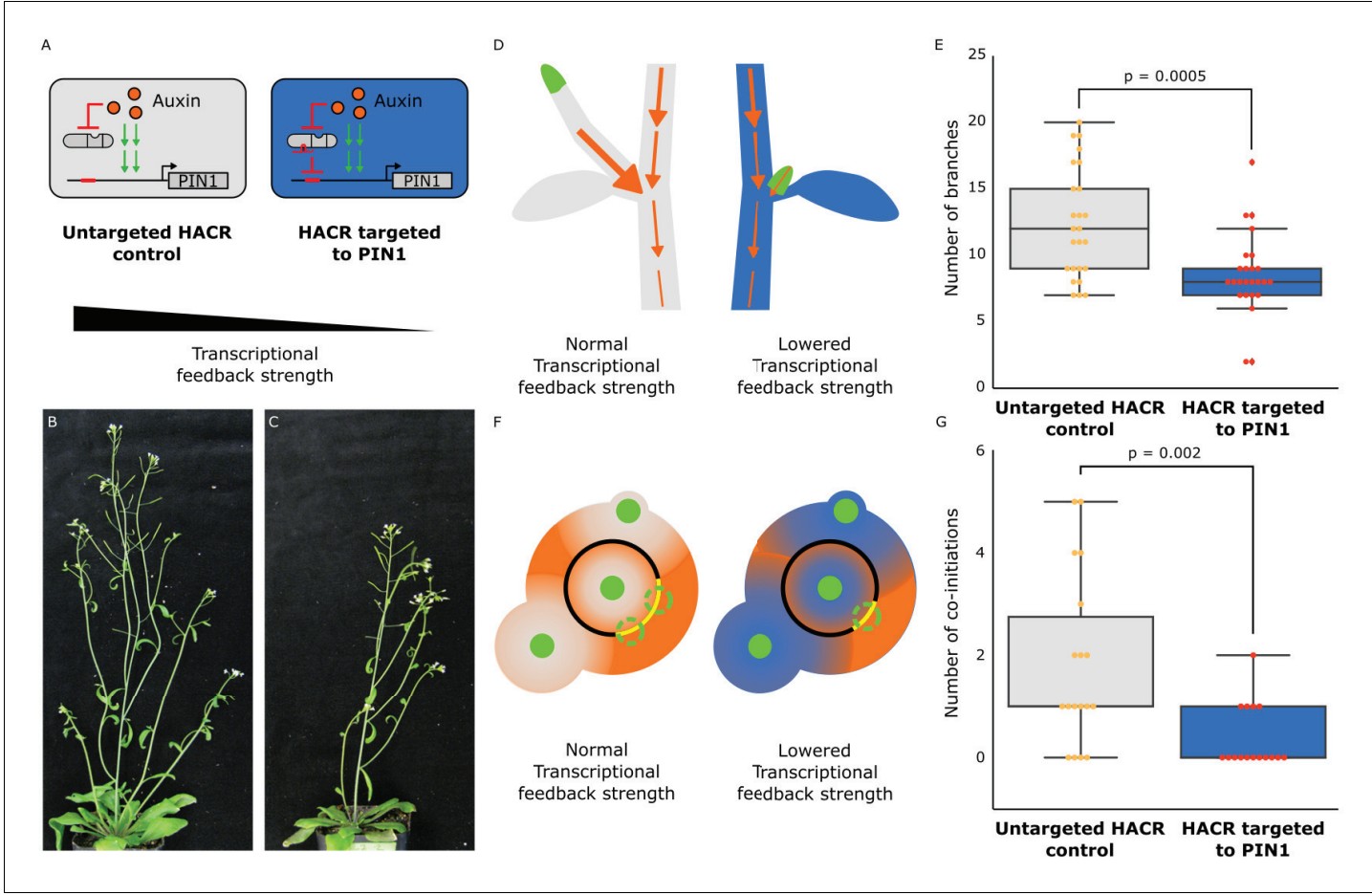

**Figure 3.** The developmental circuit regulating branching can be rewired using auxin HACRs. (**A**) Schematics of auxin driven PIN1 expression, which is one of the drivers of transcriptional feedback. In the box on the right we show how we decreased PIN1 transcriptional feedback strength by targeting an auxin HACR to regulate PIN1. (**B,C**) Representative pictures of T3 plants of the same age without (**B**) and with (**C**) a gRNA targeting an auxin HACR to regulate PIN1. (**D**) Schematic of the mechanism behind the predicted decrease in branching from decreasing transcriptional feedback strength. In plants without a HACR targeted to PIN1 (grey), the stronger transcriptional feedback allows the lateral bud (green) to drain auxin (orange arrows) into the central vasculature. In plants with a HACR targeted to PIN1 leading to reduced transcriptional feedback (blue), the bud is not able to drain its auxin, preventing branch formation. (**E**) Box plots summarizing the number of branches of adult T3 plant lines (n = 25) with a HACR targeted to regulate PIN1 expression (blue boxes), compared to control lines that did not have a gRNA targeting PIN1 (grey boxes). Every dot represents an individual plant. (**F**) Schematic depicting the role of transcriptional feedback in the pattern of formation of new primordia (green circles) around the shoot apical meristem. We hypothesize that in the shoot apex of lines without a HACR targeting PIN1 (grey) the stronger transcriptional feedback leads to smaller zones of auxin depletion around primordia compared to lines that have a HACR targeting PIN1 (blue). This leads to a broader zone where auxin can accumulate (orange) and create new primordia (dashed green circles) which increases chances of phyllotactic defects. (**G**) Box plots summarizing the number of co-initiations in T3 plant lines (n = 25) with a HACR targeted to regulate PIN1 expression (blue boxes), compared to parental control lines that did not have a gRNA targeting PIN1 (grey boxes). Every dot represents an individual plant. All p-values reported were calculated using a one-way ANOVA.
DOI: https://doi.org/10.7554/eLife.34702.014

The following source data and figure supplements are available for figure 3:

**Source data 1.** Branching data for *Figure 3*.
DOI: https://doi.org/10.7554/eLife.34702.020
**Source data 2.** Co-initiations data for *Figure 3*.
DOI: https://doi.org/10.7554/eLife.34702.021
**Source data 3.** qPCR data *Figure 3—figure supplement 1*.
DOI: https://doi.org/10.7554/eLife.34702.022
**Source data 4.** Auxin HACR Background 1 T2 branching data *Figure 3—figure supplement 2*.
DOI: https://doi.org/10.7554/eLife.34702.023
**Source data 5.** Auxin HACR Background 2 T2 branching data *Figure 3—figure supplement 2*.
DOI: https://doi.org/10.7554/eLife.34702.024

*Figure 3 continued*

**Source data 6.** Co-initiation data for *Figure 3—figure supplement 4*.
DOI: https://doi.org/10.7554/eLife.34702.025
**Source data 7.** Data for auxin HACR background 2 *Figure 3—figure supplement 5*.
DOI: https://doi.org/10.7554/eLife.34702.026
**Source data 8.** Data for auxin HACR background 3 *Figure 3—figure supplement 5*.
DOI: https://doi.org/10.7554/eLife.34702.027
**Figure supplement 1.** The effect of targeting a HACR to regulate PIN1 can be predicted using ordinary differential equations and qualitatively validated using qPCR.
DOI: https://doi.org/10.7554/eLife.34702.015
**Figure supplement 2.** The developmental circuit regulating branching can be predictably rewired using auxin HACRs.
DOI: https://doi.org/10.7554/eLife.34702.016
**Figure supplement 3.** A HACR variant with a stabilized auxin degron does not produce a reduced branching phenotype when targeted to PIN1.
DOI: https://doi.org/10.7554/eLife.34702.017
**Figure supplement 4.** The developmental circuit regulating phyllotaxy can be predictably rewired using HACRs.
DOI: https://doi.org/10.7554/eLife.34702.018
**Figure supplement 5.** The shoot architecture phenotypes generated by targeting a HACR to regulate PIN1 are not due to antibiotic selection and can be observed in multiple different lines and Auxin HACR backgrounds.
DOI: https://doi.org/10.7554/eLife.34702.019

The gRNA expression cassettes contain a sgRNA driven by the U6 promoter and have a U6 terminator. The HACR expression cassettes are driven by the constitutive UBQ10 (AT4G05320) promoter and have a NOS terminator. All HACR variants contain the same deactivated SpCas9 (dCas9) domain (*Gilbert et al., 2013*) translationally fused at the N-terminus to an SV40 nuclear localization signal. The hormone degron domain and the repressor domain were fused to the C terminus of dCas9, with the respective degron domain in the middle and flexible 6xGS linkers separating the sub-domains. The rapidly degrading NdC truncation of the IAA17 degron (*Moss et al., 2015*) was used for all the auxin HACRs described in the paper. The JA HACR contained the degron from the Arabidopsis JAZ9 protein (AT1G70700) (*Katsir et al., 2008*). The GA HACRs contained either GAI (At1g14920) (*Murase et al., 2008*) or RGA1 (At2g01570) (*Murase et al., 2008*) cloned from *Arabidopsis* cDNA. The HACR repression domain was the nucleic acid sequence corresponding to the first 300 amino acids of the TOPLESS repressor (TPL, At1g15750) (*Pierre-Jerome et al., 2014*). We chose this repression domain as TPL is the co-repressor used in native auxin and JA signal transduction pathways. The reporter cassette that was regulated by the HACRs contained a yellow fluorescent protein (Venus) translationally fused to a nuclear localization sequence on its N-terminus and firefly luciferase translationally fused on its C-terminus with flexible linkers. The reporter was driven by a constitutive UBQ1 (AT3G52590) promoter and had a UBQ1 terminator. The additional reporter in the ratiometric lines was identical to these constructs except Venus-Luciferase was replaced with tdTomato and the gRNA target site in the UBQ1 promoter was mutated. The PIN1 gRNA expression vector contained a U6 promoter and terminator.

## Construction of plant lines

All HACR reporter lines were built by transforming the yeast artificial chromosome plasmids described above into *Agrobacterium tumefaciens* (GV3101) and using the resulting strains to transform a Columbia-0 background by floral dip (*Clough and Bent, 1998*). Transformants were then selected using a light pulse selection (*Harrison et al., 2006*). Briefly, this involves exposing the seeds to light for 6 hr after stratification (4°C for 2 days in the dark) followed by a three day dark treatment. Resistant seedlings demonstrate hypocotyl elongation in the case of Hygromycin and leaf greening after 5 days in the case of Kanamycin. After selection seedlings were transplanted to soil and grown in long day conditions at 22°C.

For all the HACR reporter genotypes (*Figures 1* and *2*) at least three lines were grown to the T2 and tested for their response to the appropriate hormone treatment with n = 10 for seedlings. To generate the ratiometric auxin HACR lines the additional tdTomato reporter was transformed into Col0 and then lines that were screened for uniform tdTomato expression were crossed into a line that had the HACR targeted to a Venus reporter.

Three different auxin HACR backgrounds were transformed with a gRNA targeting PIN1. The branching of three independent lines, representing three independent PIN1 gRNA insertion events, in each HACR background was characterized in the T2 at n = 5. Several lines were characterized in the T3 at n > 20 both with and without selection. The number of co-initiations of three independent lines in one HACR background was characterized in the T2 at n = 5. The number of co-initiating siliques of one of these lines was characterized in the T3 at n = 25.

## Fluorescence microscopy

For imaging the effects of auxin treatment on root tips we selected plants on 0.5xLS +0.8% bactoagar containing Kanamycin using the light pulse protocol described above. Four days after the seedlings were removed from the dark we transplanted to fresh 0.5xLS +0.8% bactoagar without Kanamycin and then imaged on a Leica TCS SP5 II laser scanning confocal microscope on an inverted stand. For auxin induction of root tips, the seedlings were sprayed with a 1:1000 dilution in water of either control (DMSO) or auxin dissolved in DMSO (5 μM final concentration) and then mounted on slides in water and imaged after 24 hr.

For the imaging of ratiometric lines seedlings were germinated without selection and then visually screened using a fluorescence microscope for expression of both reporters. These seedlings were then imaged on a confocal microscope at several positions along the primary root to visualize auxin distributions in the root tip, the elongation zone and in developing lateral roots. The images were taken using a Leica TCS SP5 II laser scanning confocal microscope on an inverted stand. The ratiometric images were generated using the calcium imaging calculator in the Leica software, by background subtracting both the tdTomato and Venus signals and then normalizing the Venus signal by the tdTomato signal.

The images of ovules 48 hr after pollination were obtained by emasculating flowers prior to anther dehiscence followed by hand pollination 12 hr after. After 48 hr, the ovules from the pistils of these flowers were dissected using hypodermic needles under a dissection microscope and then mounted on slides in 80 mM sorbitol and imaged with confocal microscopy as in Beale et al. (*Beale et al., 2012*). To image the developing embryos, ovules were dissected from siliques at the appropriate developmental stages, individually dissected and mounted onto slides in MS0 media before being analyzed by confocal microscopy. All confocal microscopy images presented in this work are maximum projections of sub-stacks from regions of interest.

## Luciferase assays

Luciferase based time course assays were used to characterize the dynamics of HACR response to exogenous or endogenous hormone stimulus. All imaging was done using the NightOWL LB 983 in vivo Imaging System, which uses a CCD camera to visualize bioluminescence. For the data collected for *Figure 1* and *Figure 3—figure supplement 1*, assays were performed on seedlings. Here, T2 plants were selected by Kanamycin selection using the previously described light pulse protocol. These were then transplanted to fresh plates without antibiotic four days after selection and sprayed with luciferin (5 μM in water) in the evening. The next morning, after approximately 16 hr, they were sprayed again with luciferin. After 5 hr they were imaged for one hour (10 min exposure with continuous time points), then sprayed with a control treatment (a 1:1000 dilution of DMSO in water) and then imaged for five hours. These same plates were then re-sprayed with luciferin (5 μM in water) and left overnight. The next day these same plates were again imaged with an identical protocol as the previous day, except they were sprayed with a 1:1000 dilution of hormone in water (5 μM Indole-3-acetic acid (auxin), 30 μM coronatine (JA) or 100 μM GA3 post dilution) rather than control. Luminescence of each seedling was recorded over time and reported as values normalized to the time-point prior to treatment. For the mechanical damage assay of the jasmonate HACR in *Figure 2*, plants were treated identically as described above except that instead of being sprayed with hormones, leaves on the plant were mechanically crushed using forceps.

## Data analysis

All the data collected was analyzed and plotted using python (*Khakhar, 2018*; https://github.com/arjunkhakhar/HACR_Data_Analysis; *copy archived at https://github.com/elifesciences-publications/HACR_Data_Analysis). For the luciferase assays, all the time courses were normalized the reading

before induction to make them comparable. All p-values reported were calculated in python using the one-way ANOVA function from the SciPy package (*Oliphant, 2007*). (https://docs.scipy.org/doc/scipy/reference/generated/scipy.stats.f_oneway.html)

## Characterizing plant phenotypes

To characterize branching in plant lines with and without an auxin HACR regulating PIN1, we selected T2 transformants for lines that had a gRNA targeting PIN1 and the parental HACR background that had no gRNA. The plants that passed the selection were transplanted onto soil and then characterized as adults at the point that there were on average four stems on the no gRNA control lines. In all cases the parental controls that lack a gRNA and the lines derived from them, by transforming with a gRNA targeting PIN1, were all grown in parallel and phenotyped on the same day to ensure the data collected was comparable. Additionally, while we do not believe that the selection would have a significant effect on the phenotyping data as we collected it more than a month after the plants had been transplanted off selection plates onto soil, both the lines with a PIN1 targeting gRNA and the parental controls they were compared to were selected in parallel to control for any confounding effect. Phenotyping involved counting the number of branches on the plant. We quantified the number of branches on five T2 plants for three different lines with a HACR targeted to regulate PIN1 in two different HACR backgrounds, in parallel with the parental HACR background. The line with the strongest phenotype was propagated to the T3 generation with its parental HACR background and the same experiment was repeated with an n = 25. To quantify the number of co-initiating siliques we measured the internode length between the first 20 siliques on a single axillary stem and every instance of two siliques emerging from the same point on the stem (an internode length less than 1 mm which we found to be the threshold for visual discrimination) was considered a co-initiation. The line that showed the strongest phenotype was propagated to the T3 generation with its parental HACR background and the same experiment was repeated with an n = 25.

To prove the phenotypes we were observing were independent of selection conditions we also characterized branching of T2 and T3 plant lines that were not selected on antibiotic selections. These plant lines were transplanted off 0.5x LS plates ten days after germination. They were then grown till adulthood and then phenotyped and genotyped for the presence of the HACR and PIN1 gRNA.

All plants that were phenotyped were grown in long day conditions on Sunshine #4 mix soil in rose pots and watered every other day on a watering table.

## qPCR assays

All qPCR assays were performed on seedlings seven days after they been selected using the light pulse procedure (fifteen days post germination). For each biological replicate five seedlings that passed selection were transplanted off the selection plate and into 4 ml of 0.5xLS with either mock of 50 nM 2-4D. They were then incubated in well lit, humidity-controlled conditions for 3 hr and then the seedlings were blotted and flash frozen in liquid nitrogen. The RNA was extracted from these seedlings using the Illustra RNAspin Mini Kit from GE. cDNA was then prepared from 1 ug of RNA using the iScript cDNA synthesis kit from Biorad and then used to run a qPCR with the iQ SYBR Green Supermix also from Biorad on a Biorad qPCR machine. Each sample was analyzed for expression of PIN1 and PP2A which was used to normalize PIN1 levels. A standard curve was generated using the pooled samples for each primer set to determine amplification efficiency. The primers used are listed below:

PIN1_q_R: AACATAGCCATGCCTAGACC
PIN1_q_F: CGTGGAGAGGGAAGAGTTTA
PP2A_q_R: AACCGCTTGGTCGACTATCG
PP2A_q_F: AACGTGGCCAAAATGATGC

## Plant genotype list

| Plant genotype | Used in the following figure |
| --- | --- |

*Continued on next page*

*Continued*

| Plant genotype | Used in the following figure |
|---|---|
| ABS44 (p2301Y-tOCS-pUBQ1:NLS-Venus-LucPlus-tUBQ1-pU6:pUBQ1_gRNA_Target1-tU6-pUBQ10:dCas9-NdC_IAA17-TPLRD2-tNos) | *Figure 1B–F, Figure 2F–J, Figure 3B,E,G,H, Figure 1—figure supplement 1, Figure 2* |
| PHD5 (p2301Y-tOCS-pUBQ1:NLS-Venus-LucPlus-tUBQ1-pU6:pUBQ1_gRNA_Target1-tU6-pUBQ10:dCas9-Jas9-TPLRD2-tNos) | *Figure 1H, Figure 2N–Q, Figure 1—figure supplement 1, Figure 2—figure supplement* |
| PHD3 (p2301Y-tOCS-pUBQ1:NLS-Venus-LucPlus-tUBQ1-pU6:pUBQ1_gRNA_Target1-tU6-pUBQ10:dCas9-GAI1-TPLRD2-tNos) | *Figure 1J, Figure 2K–M, Figure 1—figure supplement 1, Figure 2—figure supplement* |
| PHD6 (p2301Y-tOCS-pUBQ1:NLS-Venus-LucPlus-tUBQ1-pU6:pUBQ1_gRNA_Target1-tU6-pUBQ10:dCas9-RGA1-TPLRD2-tNos) | *Figure 1I, Figure 1—figure supplement 1, Figure 2—figure supplement 1* |
| ABS44 (p2301Y-tOCS-pUBQ1:NLS-Venus-LucPlus-tUBQ1-pU6:pUBQ1_gRNA_Target1-tU6-pUBQ10:dCas9-NdC_IAA17-TPLRD2-tNos) +pGRN_H-pU6:pPIN1_gRNA_Target1-tU6 | *Figure 3C,E,G,H, Figure 1—figure supplement 1, Figure 2—figure supplement 1* |
| ABS50 (p2301Y-tOCS-pUBQ1:NLS-Venus-LucPlus-tUBQ1-pU6:pUBQ1_gRNA_Target1-tU6-pUBQ10:dCas9-IAA28_DegronDead-TPLRD2-tNos) | *Figure 1D,F, Figure 3—figure supplement 1* |
| ABS50 (p2301Y-tOCS-pUBQ1:NLS-Venus-LucPlus-tUBQ1-pU6:pUBQ1_gRNA_Target1-tU6-pUBQ10:dCas9-IAA28_DegronDead-TPLRD2-tNos) +pGRN_H-pU6:pPIN1_gRNA_Target1-tU6 | *Figure 3—figure supplement 1* |

## Plasmid maps

ABS44 - https://benchling.com/s/yXKJkba5

ABS50 - https://benchling.com/s/897tnlX2

PHD5 - https://benchling.com/s/HnODIKMV

PHD3 - https://benchling.com/s/HOEPc5FA

PHD6 - https://benchling.com/s/Ge8pztYw

pGRN_H-pU6:pPIN1_gRNA_Target1-tU6 - https://benchling.com/s/3RBYAIkF

pGRN_H-pUBQ1_AlteredGrnaTargetSite:NLS-tdTomato-tUBQ1 - https://benchling.com/s/Pd0Ms4Qs

## Acknowledgements

We thank Dr. Takato Imazumi for sharing resources and advice, particularly on luciferase imaging; Dr. Dominic Loque for materials to build the yeast artificial chromosomes ahead of publication; Ms. Mrunmayee Shete and the Aquarium team for technical assistance; Dr. Mallorie Taylor-Teeples for resources and advice related to phyllotactic patterning; and Dr. Hardik Gala for advice on microscopy. We would like to thank Mr. Randolph Lopez and Dr. Clay Wright for commenting on the manuscript, and all members of the Klavins, Seelig and Nemhauser labs for their advice and input. This work was supported by shared grants to EK and JLN from the National Science Foundation (MCB-1411949) and the National Institutes of Health (R01-GM107084), as well as support from the Howard Hughes Medical Institute Faculty Scholars Program to JLN.

## Additional information

### Funding

| Funder | Grant reference number | Author |
|---|---|---|
| National Science Foundation | MCB-1411949 | Eric Klavins |
| National Institutes of Health | R01-GM107084 | Jennifer L Nemhauser |
| Howard Hughes Medical Institute | Faculty Scholars Program | Jennifer L Nemhauser |

The funders had no role in study design, data collection and interpretation, or the decision to submit the work for publication.

## Author contributions
Arjun Khakhar, Conceptualization, Data curation, Formal analysis, Investigation, Visualization, Writing—original draft, Project administration, Writing—review and editing; Alexander R Leydon, Validation, Investigation, Methodology, Writing—original draft, Writing—review and editing; Andrew C Lemmex, Validation, Investigation, Writing—review and editing; Eric Klavins, Jennifer L Nemhauser, Conceptualization, Funding acquisition, Writing—original draft, Writing—review and editing

## Author ORCIDs
Arjun Khakhar ![ORCID] http://orcid.org/0000-0002-4676-6533
Eric Klavins ![ORCID] http://orcid.org/0000-0002-3805-5117
Jennifer L Nemhauser ![ORCID] http://orcid.org/0000-0002-8909-735X

## Decision letter and Author response
Decision letter https://doi.org/10.7554/eLife.34702.031
Author response https://doi.org/10.7554/eLife.34702.032

# Additional files

## Supplementary files
• Transparent reporting form
DOI: https://doi.org/10.7554/eLife.34702.029

## Data availability
Links to all plasmid sequences are in manuscript and data analysis code is available on github.

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

# Appendix 1

DOI: https://doi.org/10.7554/eLife.34702.030

## Supplementary materials

### Supplementary note 1

We built a model that captured the transcriptional activation of PIN1 by auxin and its repression by an auxin responsive HACR at the mRNA and protein levels. In this model PIN1 expression is activated proportional to the auxin concentration and repressed proportional to the auxinHACR concentration. Auxin causes the degradation of the auxinHACR protein. In addition to the passive diffusion of auxin in and out of the cell, auxin is actively transported out at a rate proportional to the concentration of PIN1. While the quantitative behavior of the model is dependent on the parameter set chosen, such as the repression strength, as we intend to use the model to make purely qualitative predictions all parameter values were chosen to generate biologically plausible behavior of the wildtype and have arbitrary units. The fact that the relative expression levels of PIN1 seem to agree with the qualitative predictions of model (*Figure 3—figure supplement 1*) implies that the parameter set is plausible. This model allows us to make qualitative predictions of how we would expect a HACR to perturb PIN1 expression. It also serves to highlight the significant differences that hormone responsive and static repression have on both the dynamic and steady state expression of PIN1 in response to auxin. The equations used to build the model, as well as the parameter values are listed below.

$$\varphi_{PIN1_{mRNA}} = 1$$

$$\theta_{PIN1_{mRNA}} = 1$$

$$\varphi_{AuxinHACR_{mRNA}} = 1$$

$$K_{Repression\ strength} = 10$$

$$\delta_{PIN1} = 1$$

$$\mu_{PIN1} = 0.1$$

$$\delta_{PIN1} = 2$$

$$\mu_{PIN1} = 0.1$$

$$K_{degradation\ rate} = 5$$

$$K_{Auxin\ diffusion\ in} = 1$$

$$K_{Auxin\ diffusion\ out} = 0.01$$

$$K_{PIN1\ transport\ efficiency} = 1$$

$$K_{Auxin\ activation} = 1$$

$$\frac{d[PIN1_{mRNA}]}{dt} = \varphi_{PIN1_{mRNA}}$$
$$\times \left( \frac{K_{Auxin\ activation} \times [\text{Auxin}]}{K_{Auxin\ activation} \times [\text{Auxin}] + \theta_{PIN1_{mRNA}} + K_{Repression\ strength} * [\text{AuxinHACR}]} - [\text{PIN1}_{mRNA}] \right)$$

$$\frac{d[AuxinHACR_{mRNA}]}{dt} = \varphi_{AuxinHACR_{mRNA}} \times (1 - [AuxinHACR_{mRNA}])$$

$$\frac{d[PIN1]}{dt} = \delta_{PIN1} \times [\text{PIN1}_{mRNA}] - \mu_{PIN1} \times [\text{PIN1}]$$

$$\frac{d[AuxinHACR]}{dt} = \delta_{AuxinHACR} \times [\text{AuxinHACR}_{mRNA}] - \mu_{AuxinHACR} \times [\text{AuxinHACR}]$$
$$- K_{degradation\ rate} \times [\text{Auxin}] \times [\text{AuxinHACR}]$$

$$\frac{d[Auxin]}{dt} = K_{Auxin\ diffusion\ in} - K_{Auxin\ diffusion\ out} * [\text{Auxin}] - [\text{PIN1}] \times K_{PIN1\ transport\ efficiency} \times [\text{Auxin}]$$

