## [Decision Letter]

Thank you for submitting your article "Synthetic hormone-responsive transcription factors can monitor and re-program plant development" for consideration by *eLife*. Your article has been favorably evaluated by Christian Hardtke (Senior Editor) and three reviewers, one of whom is a member of our Board of Reviewing Editors.

The reviewers have discussed the reviews with one another and the Reviewing Editor has drafted this decision to help you prepare a revised submission. Given the interesting applications of your reporter system, we think this manuscript is a better fit as a Tools and Resources paper. This in no way changes the value we place on publishing this submission, but given the nature of this contribution, we feel it is more of a technical rather than a conceptual advance.

Summary:

With the expansion of systems and modeling approaches in developmental biology (and plant biology) the need for tools to test predictions of these approaches is increasing. This paper describes a bioengineering tool that uses CRISPR-Cas9 based targeting to bring selected genes under the control of phytohormones that signal by targeted protein degradation. This provides a way to (a) tune the transcription from these genes by exogenous hormone addition; (b) report levels of endogenous hormone signaling; and (c) re-parameterize hormone signaling networks at specific nodes. The basic system has already been described in yeast, but this paper aims to provide in planta evidence of the efficacy of these three applications.

Overall, the work described is an important a proof of concept, and all three reviewers felt that the HACRs could be valuable tools for the research community. The two major concerns were about (1) the limited robustness of the phenotypic analysis, and (2) whether it was clearly demonstrated that these tools enabled manipulations not possible with existing tools. Experiments suitable to address these issues are delineated below.

Essential revisions:

1) Analysis of phenotypes should be extended to T3 plants not grown under antibiotic selection. The authors argue that it is robust, given their controls and the long time between the selection and the phenotypic analysis. However, antibiotic selection often has long term effects in the primary shoot apical meristem, which can affect both the phenotypes analysed. Therefore use of unselected homozygous T3 lines necessary for truly reliable data. Extension of the n=25 T3 approach to additional independent lines is advisable.

2) Clear evidence that addition of the targeted HACR repressor will affect the dynamics of PIN1 induction and dampen positive feedback between auxin and PIN1 levels is needed. The authors assume that addition of the targeted HACR repressor will affect the dynamics of PIN1 induction and dampen positive feedback between auxin and PIN1 levels. This is never directly tested. Ideally it would be better to use a tagged PIN1 that rescues the pin1 mutant phenotype and validate the authors hypotheses about PIN polarization in buds and meristems more directly.

Alternatively some transcription dose-response analysis could be undertaken to test whether the induction threshold for auxin really is shifted as expected.

3) An example of an experiment that could only have been done with HACRs (and not weak alleles or existing tools) would be a strong addition to this paper. This could be done alongside a transcription dose-analysis as suggested in point 2. Alternatively, taking advantage of the inducibility of the constructs to provide spatial or temporally restricted manipulations followed by phenotypic analysis.

4) More information about the predictions from existing models is necessary. It is not clear from notes 1 and 2 whether the authors ran simulations to generate these predictions or rather intuited them from examination of model terms. Because of the relationship between auxin transport and auxin concentration, it is not easy to predict what the effects of changing the relationship between auxin concentration and PIN1 expression might be. If expression of PIN1 increases proportionately PIN1 at the plasma membrane, then increasing PIN1 expression will decrease cellular auxin concentration. Because of this negative feedback of PIN1 on auxin concentration, the effects on the system of reducing the positive feedback of auxin on PIN1 may not be very intuitive, especially at a tissue level.

---

## [Author Response]

Essential revisions:

1) Analysis of phenotypes should be extended to T3 plants not grown under antibiotic selection. The authors argue that it is robust, given their controls and the long time between the selection and the phenotypic analysis. However, antibiotic selection often has long term effects in the primary shoot apical meristem, which can affect both the phenotypes analysed. Therefore use of unselected homozygous T3 lines necessary for truly reliable data. Extension of the n=25 T3 approach to additional independent lines is advisable.

While it is common in our field to phenotype plants that have undergone selection (particularly weeks after the plants have been transplanted to soil), we appreciate the reviewer’s desire for heightened scrutiny of the HACR-dependent phenotypes. Towards that end, we repeated the shoot architecture phenotyping experiments on T3 plants that were never exposed to selection (genotypes were confirmed by PCR post phenotyping). This data is summarized in Figure 3—figure supplement 5. As shown, we observed similar trends of reduced branching to those originally reported. Given the time constraints of the review process, we were not able to extend this analysis to the co-initiations. However, we believe that the branching data serves as sufficient evidence that the selection is not the cause of the shoot phenotypes we are observing.

On the reviewer’s advice we also extended the phenotyping to several new lines and a new Auxin HACR background (all analyzed without selection) to demonstrate that these phenotypes can be replicated in multiple independent transformants. This data is also summarized in Figure 3—figure supplement 5.

2) Clear evidence that addition of the targeted HACR repressor will affect the dynamics of PIN1 induction and dampen positive feedback between auxin and PIN1 levels is needed. The authors assume that addition of the targeted HACR repressor will affect the dynamics of PIN1 induction and dampen positive feedback between auxin and PIN1 levels. This is never directly tested. Ideally it would be better to use a tagged PIN1 that rescues the pin1 mutant phenotype and validate the authors hypotheses about PIN polarization in buds and meristems more directly.Alternatively some transcription dose-response analysis could be undertaken to test whether the induction threshold for auxin really is shifted as expected.

We agree with the reviewer that the evidence we provided for an auxin-dependent effect on PIN1 expression was indirect. We approached this issue in two ways. First, we built a simple ordinary differential equation-based model of PIN1 transcriptional regulation by a HACR, which is described in Supplementary note 1. This model predicted that the HACR would damp the auxin dose response relationship, as shown in Figure 3—figure supplement1. We then qualitatively validated this prediction by performing an experiment suggested by the reviewers: qPCR on plants with a HACR or with a HACR targeted to regulate PIN1 exposed to either mock or auxin treatments. We observed the predicted reduction in *PIN1* levels at endogenous auxin concentrations and a relief of repression at higher auxin concentrations. This data is summarized in Figure 3—figure supplement 1.

3) An example of an experiment that could only have been done with HACRs (and not weak alleles or existing tools) would be a strong addition to this paper. This could be done alongside a transcription dose-analysis as suggested in point 2. Alternatively, taking advantage of the inducibility of the constructs to provide spatial or temporally restricted manipulations followed by phenotypic analysis.

We thank the reviewer for pointing out a deficit in our original submission – a clear explanation of what makes the HACR unique. The capacity to reprogram the hormone-based transcriptional regulation of a specific locus *in planta* is a unique capability of HACRs. This that is not possible with weak alleles or existing tools. We have made substantial revisions to the text to make this distinction more apparent, in particular contrasting the impact of hormone-regulated vs. static repressors. In addition, the model and data in Figure 3—figure supplement 1 significantly bolsters this point (in addition to the lack of phenotype observed in a plant transformed with a non-auxin-responsive HACR). We acknowledge that the reduced branching phenotype we generated with HACRs is not unique, but for a HACR proof-of principle experiment, we needed a well-studied system where an existing model made a strong prediction. We believe that our HACR data is the first experimental test that altering the strength of positive feedback of PIN1 by auxin is sufficient to produce the shoot architecture phenotypes. We have tried to make this distinction clearer in the body of the main text by emphasizing the pitfalls current approaches and highlighting how HACRs can get around these to achieve the goal rewiring hormonal transcriptional regulation in a precise and tunable manner.

4) More information about the predictions from existing models is necessary. It is not clear from notes 1 and 2 whether the authors ran simulations to generate these predictions or rather intuited them from examination of model terms. Because of the relationship between auxin transport and auxin concentration, it is not easy to predict what the effects of changing the relationship between auxin concentration and PIN1 expression might be. If expression of PIN1 increases proportionately PIN1 at the plasma membrane, then increasing PIN1 expression will decrease cellular auxin concentration. Because of this negative feedback of PIN1 on auxin concentration, the effects on the system of reducing the positive feedback of auxin on PIN1 may not be very intuitive, especially at a tissue level.

We appreciate this feedback, and have attempted to clarify how we used existing models to guideour work. We have moved the description of the models from the supplementary notes to the main text in paragraphs 11 and 12 (as suggested by the reviewers in minor point 7 below) and substantively re-written these sections to improveclarity.